# Treatment of Chronic Hepatitis D with Bulevirtide—A Fight against Two Foes—An Update

**DOI:** 10.3390/cells11223531

**Published:** 2022-11-08

**Authors:** Peter Ferenci, Thomas Reiberger, Mathias Jachs

**Affiliations:** Division of Gastroenterology and Hepatology, Department of Internal Medicine III, Medical University of Vienna, 1090 Vienna, Austria

**Keywords:** treatment hepatitis D, hepatitis B, bulevirtide, interferon-alfa

## Abstract

HDV infection frequently causes progression to cirrhosis and hepatocellular carcinoma (HCC). In summer 2020, the first potentially effective drug Bulevirtide (BLV) has been approved for the treatment of HDV by the EMA. BLV is a synthetic N-acylated pre-S1 lipopeptide that blocks the binding of HBsAg-enveloped particles to the sodium taurocholate co-transporting polypeptide (NTCP), which is the cell entry receptor for both HBV and HDV. In this review, we discuss the available data from the ongoing clinical trials and from “real world series”. Clinical trials and real-world experiences demonstrated that BLV 2 mg administered for 24 or 48 weeks as monotherapy or combined with pegIFNα reduces HDV viremia and normalizes ALT levels in a large proportion of patients. The combination of BLV and pegIFNα shows a synergistic on-treatment effect compared with either one of the monotherapies.

## 1. Introduction

The hepatitis D virus (HDV; Delta virus, family: Komlioviridae [1]) depends on hepatitis B virus (HBV) envelope (HBV surface antigen, HBsAg) to form infectious HDV particles. Thirteen percent of carriers of the hepatitis B virus (HBV) are co-infected with HDV [2,3], resulting in an estimated number of >11 million HDV patients worldwide based on the reported global HBsAg prevalence of 3.9% [2]. HDV infection frequently causes progression to cirrhosis and hepatocellular carcinoma (HCC) [2]. Recent epidemiologic data from Austria suggest that more than half of HDV patients develop advanced chronic liver disease (ACLD), experience liver-related morbidity or require liver transplantation [4].

To date, no pharmacological therapy is able to cure chronic hepatitis D. Treatment with interferon [5] or pegylated interferon alpha (PEGIFN) [6,7] achieves the sustained suppression of HDV replication in only 25% of patients [8]. Nucleos(t)ide analogs used for the treatment of HBV do not have antiviral activity against HDV [5]. Therefore, novel HDV therapies are urgently needed.

Bulevirtide (BLV) is a synthetic N-acylated pre-S1 lipopeptide that blocks the binding of HBsAg-enveloped particles to the sodium taurocholate co-transporting polypeptide (NTCP), which is the cell entry receptor for both HBV and HDV. The liver tropism of HBV and HDV is primarily determined by a specific interaction of an extended receptor binding domain (RBD) in the pre-S1-part of the HBV L-protein [9,10] and the hepatic NTCP receptor. Therefore, BLV prevents the entry of HDV into hepatocytes, thereby decreasing the number of HDV-RNA templates needed for the synthesis of L- and S-HDAg [11,12].

## 2. Antiviral Efficacy of BLV in Patients with Chronic Hepatitis D

Based on favorable virologic efficacy and safety data of the MYR 202 and 203 studies [13], BLV has been approved for the treatment of HDV by the EMA in 2020. Unfortunately, these small phase 2 studies (see Table 1) did not provide a clear guidance for the BLV treatment of HDV patients in clinical practice. The optimal dose and duration of BLV therapy, as well the need to combine BLV with PEGIFN and/or NUC remained unclear.

Furthermore, an accepted virological efficacy surrogate marker is lacking for the BLV treatment. Currently, the combination of a > 2 log decline in viral load and normalization of ALT (combined response) is used as a primary endpoint in clinical studies. As recently outlined [18,19], the use of this endpoint implies that many patients still have detectable HDV-RNA and the impact of incomplete viral suppression on the further course of chronic hepatitis D is unclear. It is suggested that a > 2 log decline in HDV-RNA on BLV treatment defines a partial virologic response, while sustained HDV-RNA undetectability defines complete virologic response [18,19]. In addition, ALT is an uncertain marker of liver disease severity, and importantly, HDV patients with advanced chronic liver disease may have ALT in the normal range [20].

## 3. Guidance on How to Use BLV in Patients with Chronic Hepatitis D

In the absence of published studies, treatment with BLV resembles a “learning- by-doing” approach. The efficacy of HDV-directed therapies should be measured by their ability to achieve sustained suppression of HDV replication. At present, it remains unknown whether this can be achieved by BLV monotherapy. Combination therapy with other anti-viral agents may be necessary. Currently, we are awaiting the final data of the phase 2b and phase 3 data of three out of the four randomized controlled prospective studies (Table 1). These studies explore the role of treatment duration, selection of the optimal dose of BLV, and the need to combine BLV with PEGIFN. Based on the published interim date of these studies and some “real-world” studies [14,17,21,22], there is an impression that the addition of PEGIFN and/or using higher BLV doses result in higher virologic response rates.

Second, the optimal dose of BLV is uncertain; the market authorization has only been obtained for the 2 mg/d dose, while higher BLV doses may be more effective [14]. In our study [23], 20 of 21 patients who received the initial dose of 2 mg/d had some virological response (any log decline) at week 24. One patient did not sufficiently respond even after increasing the dose to 10 mg/day.

The efficacy criteria for stopping the BLV treatment are not well-defined. Treatment was terminated under close clinical monitoring in four patients who were HDV-RNA undetectable for at least 6 months [23]. All the patients relapsed but responded to the re-treatment (ongoing). One relapse occurred after >1 year of BLV cessation. This is likely due to persisting HBsAg levels that allow for the formation of new HDV particles. This finding is novel and clinically relevant, since no long-term follow-up data have been reported from previous studies using the treatment with BLV [9,10,24]. In the only study with a complete follow-up [14], only three of eighty-seven patients had undetectable HDV-RNA at the end of 24 weeks of treatment. These three patients did not relapse during 24 weeks of follow-up.

In the BLV monotherapy, HDV-RNA declined in most patients, but then levelled off and patients plateaued at a lower level. When the plateau became evident, we added PEGIFN-alfa2a, if it was not contra-indicated. Interferon-alfa2a is an accepted treatment for chronic hepatitis B. PEGIFNs have a direct antiviral efficacy against HDV [5,6]. To date, all the ongoing studies with BLV use PEGIFN-alfa2a. Alternatively, PEGIFN-lambda, which is supposed to have fewer side effects than PEGIFN-alfa2a could be used. PEGIFN-λ treatment for HBeAg positive hepatitis B showed greater early effects on HBV-DNA and qHBsAg, and comparable serologic/virologic responses at the end-of-treatment. However, post-treatment, PEGIFN-alfa associated HBeAg seroconversion rates were higher, and key secondary results mostly favored alfa [25]. More ALT flares and bilirubin elevations were observed with PEGIFN-λ. Therefore, PEGIFN-λ was not further studied in chronic hepatitis B. Currently, it is explored in combination with lonafarnib/ritonavir in chronic hepatitis D [26].

A new hope is the combination of the nucleic acid polymer (NAP) REP 2139 with the existing treatment options [27]. REP 2139 inhibits HDV-RNA replication and HDV-RNP formation via direct interaction with HDAg and HDV envelopment and release by blocking the assembly of HBV subviral particles [27].

## 4. BLV in Patients with Advanced Chronic Liver Disease (Cirrhosis)

Owing to the aggressive course of untreated HDV infection, the proportion of HDV-infected patients with advanced chronic liver disease is considerable. In an Austrian survey, almost 50% of patients that were in continuous care at one of the participating sites had severe fibrosis or cirrhosis, and were thus at high risk of developing complications of cirrhosis driven by portal hypertension [4]. Therefore, HDV patients urgently need effective antiviral therapies to prevent cirrhosis and associated complications, and to reduce the need for liver transplantation. However, prior to the advent and approval of the entry inhibitor BLV, many patients were ineligible for the treatment targeting HDV, as PEGIFN is associated with an increased risk in flares and decompensation in advanced disease, and only in patients with chronic hepatitis B and well compensated cirrhosis, can it be safely used as per the current guidelines [24].

In contrast to PEGIFN, the entry inhibitor BLV is not expected to exert deleterious effects in patients with advanced cirrhosis with or without portal hypertension. Accordingly, cirrhotic patients were included in the registration trials, most importantly in the completed and already published phase 2 MYR202 study [14] and in the phase 3 MYR301 trial, of which an interim analysis was reported at the International Liver Congress in June 2022 [16]. Both studies were conducted in cohorts that included 50% cirrhotic patients. The overall virologic response rate in the MYR301 study was 71% after 48 weeks of BLV 2 mg monotherapy, and no significant differences in antiviral efficacy were observed when stratifying the cohort by the absence or presence of cirrhosis. Furthermore, a pooled efficacy analysis of two phase 2 (MYR202, MYR203) trails and a phase 3 trial (MYR301) showed that efficacy, i.e., combined response rates after 24 weeks of treatment with 2 mg BLV per day, was similar among patients with and without cirrhosis [28]. Therefore, data from controlled clinical trials point toward a stable efficacy of the novel treatment regardless of the presence or absence of cirrhosis.

The high efficacy of BLV monotherapy was recently corroborated by data from “real-world” studies in Austria [23], Italy [29], and France [21]. The cohorts from France (*n* = 41 completed 48 weeks of BLV 2 mg monotherapy) and Austria (*n* = 19 completed 48 weeks of BLV 2 mg monotherapy) included approximately two thirds of cirrhotic patients, who in part had already developed signs of (severe) portal hypertension. The Italian cohort consisted exclusively of cirrhotic patients who had signs of portal hypertension (*n* = 18 completed 48 weeks of BLV 2 mg monotherapy). The virologic response rates were 58%, 68%, and 78% at week 48 of BLV 2 mg monotherapy in the Austrian, French, and Italian cohorts, respectively, and were thus all within the range of the phase 3 MYR301 study efficacy endpoint data. Intriguingly, suppression below the detectability threshold for HDV-RNA (which was different between the studies) was achieved in 26% (Austrian cohort), 39% (French cohort), and 23% (Italian cohort) of patients after 48 weeks of BLV 2 mg monotherapy, consistently surpassing the 12% reported in the MYR301 study.

As outlined previously, the addition of PEGIFN in patients with advanced cirrhosis should be scrutinized. In one Austrian patient who only partially responded to 2 mg BLV, ascites and jaundice developed after adding 45 µg PEGIFN/week and is currently on the transplant waiting list.

Importantly, the prognostic value of achieving a virologic/combined response to therapy remains to be demonstrated in the population of patients with advanced cirrhosis and (clinically significant) portal hypertension. One ideal surrogate endpoint for prognostic impact might be the longitudinal invasive measurement of the hepatic venous pressure gradient (HVPG), which adequately reflects portal pressure in cirrhotic patients. It has previously been demonstrated in HCV that the achievement of sustained virological response (SVR) ameliorates portal hypertension [30], and in the long run is paralleled by a drastic improvement in prognosis [31,32]. Based on these insightful studies, data on the impact of HDV-targeted therapy on portal pressure are eagerly awaited. Paired HVPG measurements were conducted in a small number of Austrian patients (*n* = 5) [23], and the achievement of virologic response was associated with regression of portal pressure in two patients who had clinically significant portal hypertension at baseline, and a stabilization of disease in one patient who had subclinical portal hypertension at baseline and did not progress during the treatment. However, these promising observations remain to be confirmed in larger network studies. Of note, non-invasive tests for the assessment of portal pressure following etiological therapy have only been validated for HCV [33] and lack validation in HBV and HDV. Therefore, we argue that based on the association of virologic and HVPG response, the efficacy of therapies targeting HDV in patients with portal hypertension should be measured by their impact on invasively measured portal pressure.

## 5. Role of Chronic Hepatitis B

In addition to the spreading of enveloped HDV via an NTCP-receptor-dependent de novo infection pathway, another mode of HDV-RNA dissemination has been described [34]. Moreover, in addition to HBV-dependent *de novo* infection, HDV propagates through cell division. It is characterized by the direct transfer of replication competent HDV-RNA between cells during mitosis [35]. IFN response efficiently suppresses cell division-mediated HDV spread. The effect of IFN treatment is more efficient when the HDV-induced IFN response is low. IFN response destabilizes HDV-RNA during cell division. This process can proceed in the absence of HBV replication. HBsAg can be derived from integrated HBV-DNA fragments into the genome of hepatocytes resulting in “empty” viral particles. This process may explain why even after reaching HDV undetectability viral relapse occurs. In our study, HBV status was studied by the repeated measurement of quantitative HBsAg levels (qHBsAg), which is the only non-invasive tool to detect changes in HBV-infected subjects treated by polymerase inhibitors [36]. Decline in qHBsAg may herald HBV cure within the next years [37]. In our study, HBsAg levels did not change during BLV monotherapy, but fell in two patients with PEGIFN add-on qHBsAg by 44% and 112%, respectively, after starting PEGINF, as shown in Figure 1. Larger decreases (< 1 log) in qHBsAg may be a favorable prognostic sign; however, it fell in two patients with PEGIFN add-on qHBsAg by 44% and 112%, respectively, after starting PEGINF. Larger decreases (< 1 log) in qHBsAg may be a favorable prognostic sign [38].

## 6. Safety

BLV monotherapy was well-tolerated even in patients with HDV-related compensated cirrhosis, who received up to 10 mg of BLV per day over 120 weeks [29]. The only observed laboratory side effect was a significant but clinically not relevant increase in serum bile acid levels during the BLV treatment, which is due to the inhibition of NTCP-mediated clearance of bile acids from the systemic circulation [8,25].

Rare cases of hypersensitivity reactions followed the administration of BLV. One patient presented with erythematous, subcutaneous plaques at the injection sides, showing a rapid increase in size up to 7 days after the injection 2 months after starting BLV (2 mg/d). The lesions were accompanied by intense pruritus. After desensitization, the treatment was restarted without any further problems [39]. In another patient, a late onset of local, T-cell-mediated allergic skin reactions were observed after BLV injection. Symptoms improved despite the continued treatment [40].

In summary, BLV is a new treatment option for chronic hepatitis D. Clinical trials and real-world experiences demonstrated that BLV 2 mg administered for 24 or 48 weeks as monotherapy or combined with pegIFNα reduces HDV viremia and normalizes ALT levels in a large proportion of patients. The combination of BLV and pegIFNα shows a synergistic on-treatment effect compared to either one of the monotherapies. Treatment is generally well-tolerated. Preliminary data of real-life studies support BLV 2 mg as long-term monotherapy even in patients with advanced compensated cirrhosis or combined with pegIFNα. However, the optimal dose as monotherapy (2 vs. 10 mg per day), the duration of therapy, stopping rules, and long-term results after stopping the treatment (sustained or maintained virologic response) must be studied.

## Figures and Tables

**Figure 1 cells-11-03531-f001:**
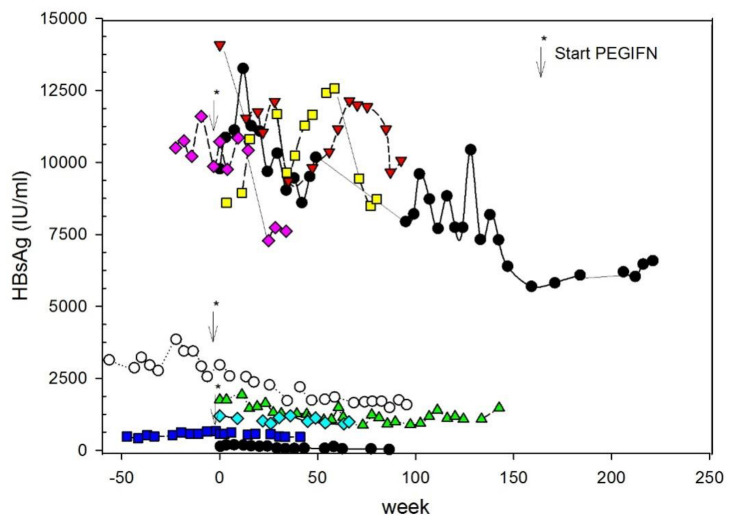
Serial measurement of qHBsAg on the treatment with Bulevirtide. In some patients, PEGIFN-alfa2a was added (the start is shown by the arrow) [23].

**Table 1 cells-11-03531-t001:** Randomized, controlled studies with Bulevirtide in chronic hepatitis D.

Study	Phase	Treatment	Duration	End of Tx Virologic Response */**	Follow-Up
Myr202 [14]	2	BLV 2 mg + TDF vs. BLV 5 mg + TDF vs. BLV10 mg + TDF vs. TDF	24	54% 55% 77% 0	Not reported
Myr203 [13]	2	BLV 5 mg BID + TDF vs. BLV10 mg + PEGIFN	48	40% 87%	Not reported
Myr204 [15] **	2b	PEGIFN (48 weeks) BLV 2 mg (96 weeks) + PEGIFN (48 weeks) BLV 10 mg (96 weeks) + PEGIFN (48 weeks) BLV 10 mg (96 weeks)	48 96 96 96	ongoing	ongoing
Myr301 [16] **	3	BLV 10 mg (48 weeks) BLV 2 mg (144 weeks) BLV 10 mg (144 weeks)	48 144 144	ongoing	ongoing

* Defined as > 2 log drop or undetectable HDV-RNA. ** For interim data after 24 or 48 weeks of treatment, see [17].

## Data Availability

Original data shown in this manuscript will be made available by the corresponding author upon reasonable request.

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
