# Peer review of "Treatment of Chronic Hepatitis D with Bulevirtide—A Fight against Two Foes—An Update"

_cells, 2022, doi:10.3390/cells11223531_

Round 1
Reviewer 1 Report
This is a well written relevant review at the time of HBV and HDV cure giving a sound understanding of the use of Bulevertide. My only point would be to include a small section regarding expected changes in the liver (speculation) with this drug.
Author Response
Sorry at the moment there are no data. Wedemaywr has shown a decline in HDV-Ag in the liver, we have shown some changes in hemodynamic parameters
Reviewer 2 Report
The paper focus on treatment options fo HDV at present and (hopefully) in the next future. The paper is well written and the theme is very actual.
Major comment:
-paragraph "role of chronic hepatitis B" could be implemented: do you suggest qHBsAg testing during treatment with BLV? are there some data on qHBsAg and HDV during treatment with pegIFN?
Minor comments:
-references:
--please check lines 27-28 as there are some references with different styles and I'm not sure citations in brackets to which reference they correspond.
--line 36-38 : add citation
--lines 119-120: check citation
-typo errors: there are some typo errors, i.e. lines 33, 63-65, 86-87, 88, 113-114
Author Response
paragraph "role of chronic hepatitis B" could be implemented: do you suggest qHBsAg testing during treatment with BLV? are there some data on qHBsAg and HDV during treatment with pegIFN?
This is already delt with in the original text:
Decline in qHBsAg may herald HBV cure within the next years (xxxvi). In 180
our study, HBsAg levels did not change during BLV monotherapy, but fell in 2 patients with PEGIFN add-on qHBsAg by 44 and112 %, respectively, after starting PEGINF. Larger decreases (<1log) in qHBsAg may be a favorable prognostic sign....
References were correct in the wordfile, in PDF some ref. # were changed to roman numbers Please keep our numbering
typos - as far as I found the, corrected